# Computer System for the Capture and Preparation of Cytopathological Reports for Cervical Cancer Detection and His Utility in Training for Health Personnel

**Sandra Olimpia Gutiérrez-Enríquez** [1] , **Miriam Celeste Guerrero-Zacarías** [1] **, Cuauhtémoc Oros-Ovalle** [2] , **Yolanda Terán-Figueroa** [1,*] **and Jorge Martín Acuña-Aradillas** [3]

1   Faculty of Nursing and Nutrition, Autonomous University of San Luis Potosi, San Luis Potosi 78290, Mexico
2   Department of Pathological Anatomy, Central Hospital "Dr. Ignacio Morones Prieto",
    San Luis Potosi 78290, Mexico
3   Smart Design Company, San Luis Potosi 78270, Mexico
*   Correspondence: yolandat@uaslp.mx; Tel.: +52-(444)-8-26-23-00 (ext. 5026)

**Abstract:** Health information systems and training are tools that support process management. The current study describes the results of the implementation of technological innovation in the process of the capture and preparation of cytopathological reports. The electronic system was structured based on national standards regarding cervical cancer control. PHP was used to design the software and MYSQL was used for the structure of the database. The total number of health personnel assigned to the cytology department participated, along with a pathologist, who made the records of the patients who came for cervical cytology to a university health center in San Luis Potosi, Mexico. The system was evaluated based on the indicators of structure, process, and results. Structure: comply with the official Mexican regulations for the registration of cervical cancer and electronic health information systems. Process: all records were legible and accurate, with varying percentages of completeness in the patient identification sections (46%) and alternate contact data (80%). Result: percentages above 80% were obtained in the satisfaction of the professionals who used the system. The system was effective as it yielded readable and accurate data that made the process of information capture and delivery of cervical screening results more efficient and faster.

**Keywords:** uterine neoplasms; PAP test; pathology; technological development; information systems; electronic health records

## 1. Introduction

Cervical cancer (CC), within the current epidemiological transition scenario, constitutes the third most common neoplasm in women worldwide [1] and the second cause of death in women in Mexico [2,3]. For this reason, and because of the great economic impact on society, it is necessary to implement control strategies through programs of prevention, detection, and effective treatment that have homogeneous information [4,5].

In order to monitor and control cases, some countries have organized statistical data; however, the quality of the information is usually variable. Faced with this situation, the World Health Organization (WHO) and the Pan American Health Organization (PAHO) suggest strengthening and innovating Health Information Systems (HIS) using new technological tools, which contribute to the management of public policies [6,7]. An example of this is the use of automated systems currently in North American and European countries, both for reading and interpreting cervical cytology samples as computer-assisted diagnostic tools, and for recording the clinical data of patients and follow-up of positive cases. With the support of these systems, it is possible to reduce human error [8,9].

The Cervical Cancer Timely Detection Program (CCTDP) in Mexico poses one of the biggest challenges, the shift toward quality culture in screening processes, from the taking

of cytology, the processing, interpretation, and delivery of results, through innovation actions and the unification of information systems [10].

In the institutions of the National Health System of Mexico, the manual format for requesting and reporting cervical cytology results is used as a reference, where information is gathered that in turn is concentrated by captors in the Women's Cancer Information System (WCIS), a database for statistical purposes and restricted access for those who are directly responsible for the care process [11]. This program is mostly executed by nursing professionals in first-level units, who document care in multiple paper formats or databases without interconnection to a pathology department, which causes more time to be invested in performing the procedure and processing the information, duplication of data, as well as a long time in the delivery of results [12].

To form efficient HIS, it is essential that the records generated from health care are of quality, as established by the Official Mexican Standard NOM-004-SSA3-2012, of the clinical record [11,13]. Since they give evidence of the care provided and are a means of communication and coordination among health personnel, they must be clear, legible, and precise. Recently it has been found the need to incorporate strategies to increase the quality of the records, and that, together with the information and communication systems, contribute to the fulfillment of goals and objectives set before the problem of morbidity and mortality due to CC in the world [14–16].

Proof of this is the results of some studies carried out in San Luis Potosí [17–19], in which the need to intervene in the factors that affect the quality of the CCTDP program records was found, including the structure of health information systems, quality control of procedures, and continuous training of human resources, as inaccurate and illegible records were observed, which may decrease the concordance between the diagnoses issued by the cytotechnologist and the pathologist [20].

Derived from the above and by the considerable importance that information systems represent in conjunction with the guidance services to improve care and facilitate effective management in administrative processes and programs, technological development was carried out for the innovation of processes. The purpose was to carry out all the clinical records of the screening tests for the timely detection of cervical cancer electronically, through software with interoperability and interconnection to a pathology department, in order to increase the efficiency of the processes of sampling, analysis, and interpretation of results. Some of the benefits that constitute an innovation as opposed to the conventional process are the following: It reduces the number of formats for collecting information, generates quality information with complete, legible, and accurate data; avoids duplication of data, speeds up the delivery of results, reducing the time of sending, processing, and analyzing data; allows access to records and statistical information for specialized research; improves communication between operational and managerial staff; improves administrative and institutional decision-making processes; and promotes sustainability environments with the elimination of materials that contribute to environmental pollution, such as paper [21].

This article describes the results of the implementation of an electronic system for the registration of cervical cytology data and cytopathology reports using the Data Capture System for Early Detection of Cervical Cancer (DCS-TDCC).

## 2. Materials and Methods

This study involved a technological innovation project, developed from the systematic use of knowledge and research, aimed to improve the process of data recording in the taking of cervical cytology and cytopathological reports, which was implemented in a health center dependent on the University of San Luis Potosí S.L.P. Mexico.

### 2.1. System Design and Operation

The DCS-TDCC was designed concerning the methodology "Technology Readiness Level (TRL)" of the National Aeronautics and Space Administration (NASA) for being a

measurement system used to assess the level of technological maturity, to which a TRL rating is assigned according to its progress, where TRL 1 is the lowest level, and TRL 9 is the highest [22].

The structure of the system contemplates the criteria of the Official Mexican Standards NOM-014-SSA2-1994 for the prevention, detection, diagnosis, treatment, control, and epidemiological surveillance of uterine cervical cancer (NOM-014-SSA2-1994) [11], NOM-024-SSA3-2010, which establishes the functional objectives and functionalities that must be observed by the products of Electronic Clinical Record Systems to guarantee the interoperability, processing, interpretation, confidentiality, security, and use of standards and catalogs of the information of the electronic health records (NOM-024-SSA3-2010) [23], and NOM-024-SSA3-2012, Electronic Health Record Information Systems. Used in the study were the Health Information Exchange (NOM-024-SSA3-2012) [24], as well as the Manual Request and Report of Results of Cervical Cytology of the Ministry of Health, consisting of four sections: (I) Identification of the Unit, (II) Identification of the Patient, (III) Background, and (IV) Result of Cervical Cytology. To design the software, PHP (Hypertext Pre-processor) was used; for the structure of the database, MYSQL (My Structured Query Language) was used; and for the server installation, the operating system UBUNTU SERVER 14.04 LTS was chosen, with processor Intel Xeon of 4GB RAM and HARD DISK 500, which was installed with Apache Server SSL (Secure Sockets Layer), SSH (Secure Shel), VIRTUAL HOST. For data security, encryption was used MD5 (Message-Digest Algorithm 5), SHA1 (Secure Hash Algorithm 1), and SALT (random seed). An internet page was enabled for participants to access the system.

The pre-analytical phase includes the care process, obtaining a sample for cytology, and the request for analysis in the Screening Department; the system provides a series of screens to record identification data, gyneco-obstetric history, and the current situation of the patient, as well as observations during gynecological examination. Upon completing the analysis request, it is sent through the DCS-TDCC to the Department of Pathology, where the analytical phase that includes the processing of the sample continues. For this purpose, the system allows the person in charge of the analysis to visualize a screen with the previously registered information to integrate it with the results of the histopathological analysis.

Once the processing is finished, the post-analytical phase continues, which includes obtaining and validating results, preparing and issuing the report, as well as interpreting it. For this, the system has a screen to capture the cytopathological result based on the Bethesda classification to generate an electronic report that can be consulted by the users involved for timely delivery to the patient.

### 2.2. Participants

The total number of health personnel assigned to the Screening Department of the health center (4) and a certified pathologist from a hospital participated and performed the records of 50 patients who came for cervical cytology during the system test time (1 March 2018 to 31 August 2019).

### 2.3. Training for System Execution

Previous training was given to the participants, divided into two phases: a theoretical one with slides and nine tutorial videos where the way of executing the system in each of its sections is explained step by step to discharge a patient and record data derived from care for cervical cytology as well as those corresponding to the analysis and interpretation of the sample and the training phase, to allow the acquisition of skills in the management of DCS-TDCC.

### 2.4. Instruments for System Testing

The effectiveness of the electronic system was evaluated in terms of structure, process, and results, with the support of three instruments, developed ex profound validated by

experts. The structure was evaluated with a ten-section checklist based on the criteria established in NOM-024-SSA3-2010 [23] and NOM-024-SSA3-2012 [24] to verify functionality through authentication; request for diagnostic assistants; clinical communication support; capture, administration, and review of clinical information; administration of patient demographics; problem list management; registration, update, and administration of the patient's medical history; patient directory; measurements, monitoring, and analysis; and finally, the report generation.

Included in the process evaluation were the completeness (record of all data), legibility (clear and understandable data), and accuracy (necessary and accurate data) of the records made by health personnel during the process of care for patients who were assigned to the module of prevention of cancer of women in the implementation period.

A checklist with 56 items was used based on the information requested by the electronic system in its sections: (I) Patient identification, (II) Alternate contact data, (III) Unit identification, (IV) Background, (V) Sampling, and (VI) A scan is observed.

The result was evaluated through a satisfaction survey of health personnel with the use of DCS-TDCC, which is composed of 21 items with a Likert scale ranging from 1 "Not satisfied", to 5 "Very satisfied".

### 2.5. System Implementation

The electronic system was restructured, and its functionality tested in terms of access, interconnection, and interoperability in the computing equipment of the Screening and Pathology Departments; once verified, each user was assigned a user ID and password to access the system and make the records electronically of the Application and Report of Results of Cervical Cytology, with storage in the database of the DCS-TDCC.

Because the sheet of paper has been deleted for registration, the system generates a sheet to identify the slide where the spreading is done, which, once fixed and dried, was packed in a slide box for shipment to the Department of Pathology, where the sample was analyzed and interpreted by the pathologist, who generated a report in the system to deliver to the patients and/or for consultation of the nursing staff of the Screening Department.

The implementation process was as follows: patient reception, interrogation, the label of the slide with folio generated by the system, sampling, gynecological assessment record, request for analysis, sending samples to pathology for analysis, and subsequent results report through the system. The interpretation of the data was carried out online by a certified pathologist, and subsequently, the staff of the Screening Department consulted these results in the system.

### 2.6. Data Processing

The structure of the system was analyzed according to the official Mexican standards mentioned above. The data of the registration process and the results in terms of satisfaction were processed in the package IBM® SPSS® Statistics 21, using descriptive statistics with simple frequency distribution, and the results were described through tables and figures.

## 3. Results

The DCS-TDCC adheres to the criteria set by the Official Mexican Standards NOM-024-SSA3-2010 and NOM-024-SSA3-2012, in compliance with the requirements for functions that must be observed in an electronic health record information system (Table 1).

The DCS-TDCC shows a home screen asking for a username and password. The main menu contains three boxes for general patient data, four boxes for the process of taking cervical cytology, four boxes for the pathology report, and one box for exiting the system. It allows for the registration of additional data and indicates the area of lesion observed during the gynecological evaluation (Figure 1).

**Table 1.** Compliance evaluation in the structure of the DCS-TDCC for recording cervical cytology data.

| Functionality Criteria | Compliance Description |
|---|---|
| Authentication. | The system requests a username and password. |
| Application for diagnostic assistants. | Allows the request for analysis by the pathologist. |
| Clinical communication support. | Maintains communication flows between personnel involved in the cervical screening process. |
| Capture, administer, and review clinical information. | Create a unique record for each patient with the option to update the information. |
| Patient demographic data management. | It stores the identification and demographic data of each patient, with the possibility of identifying it during any interaction in the care process. |
| Problem list management. | Displays diagnostic reports to follow up on patients. |
| Registration, update, and administration of the patient's medical history. | Validates the filling of mandatory minimum information, and provides updated information. |
| Patient Directory. | It has a database of identification and location of patients. |
| Measurements, monitoring, and analysis. | Complies with the Official Mexican Standards regarding health information and CC. |
| Reports generation. | It allows for the creation of electronic reports for the clinical and administrative decision-making process. |

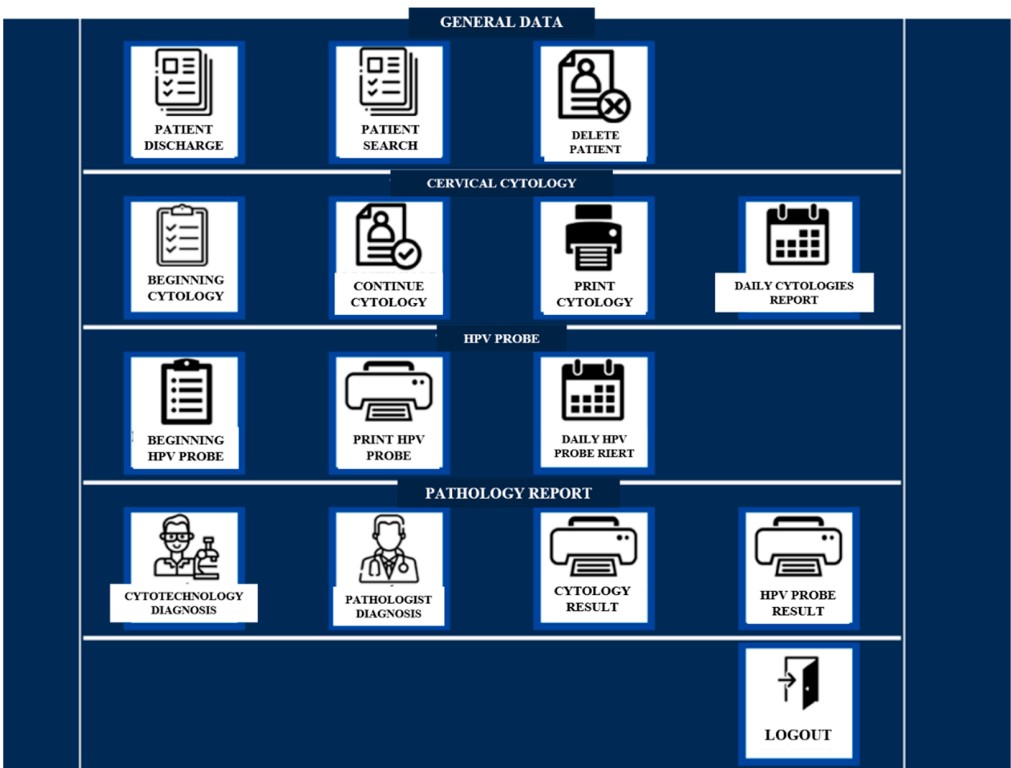

**Figure 1.** Main menu of DCS-TDCC.

As an example, Figure 2 shows the contents of the "Beginning cytology" box. As can be seen, each patient is registered with their Unique Population Registry Code (CURP for its acronym in Spanish), a code with which Mexican citizens are officially identified. Next, the identification of the health unit where the sample for cervical cytology will be taken must be registered. There you will have to register the name of the institution, which is selected from a menu. The date, entity, health jurisdiction, municipality, and medical unit are selected through different menus. There is also a specific space for recording the name of the health personnel who will take the sample and their Federal Taxpayer Registry (RFC for its acronym in Spanish).

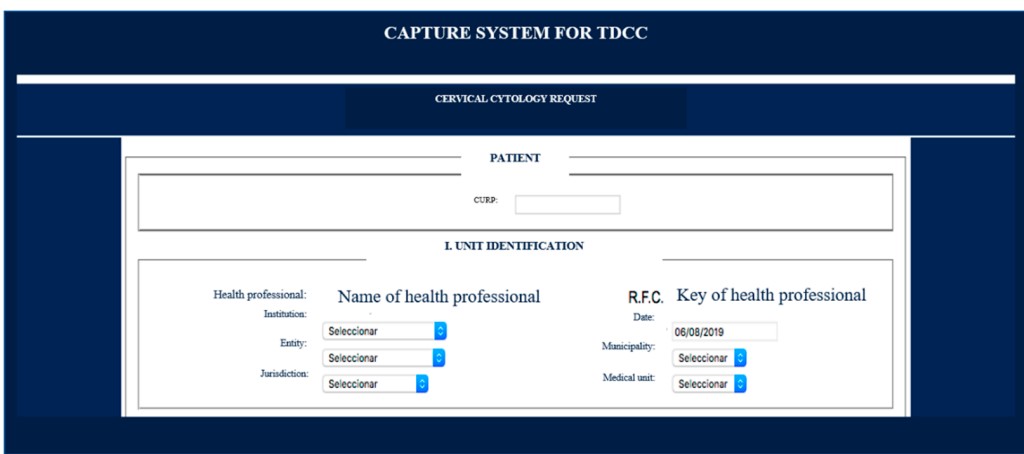

**Figure 2.** An example of cervical cytology reporting. "Beginning cytology" box.

Before the implementation of the electronic system, conventionally, 13 steps were executed with 9 different documents for the application process and reporting of cervical cytology results. With the use of the DCS-T and DCC, it was possible to reduce the number of documents used to 4 (Figure 3).

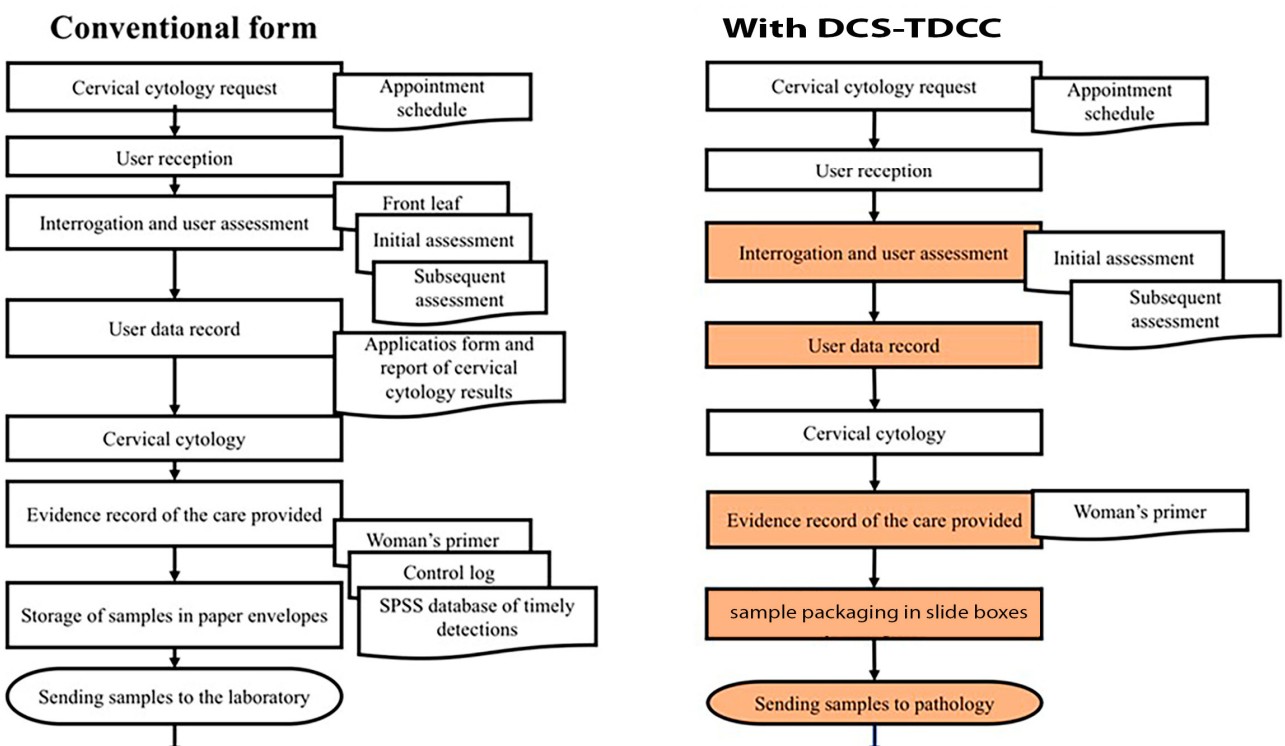

**Figure 3.** *Cont.*

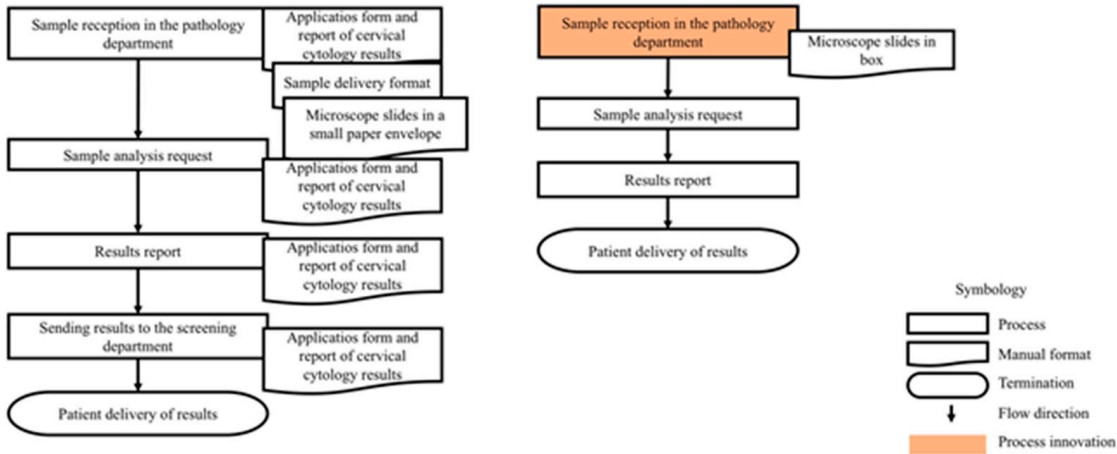

**Figure 3.** Application process and report of cervical cytology results before and during the implementation of DCS-TDCC.

The results of the evaluation show that all the records were legible and accurate; however, in the completeness, there are variable percentages in each of the sections with lower compliance in "Patient Identification" (80%) and "Data of alternate contact" (46%) (Figure 4).

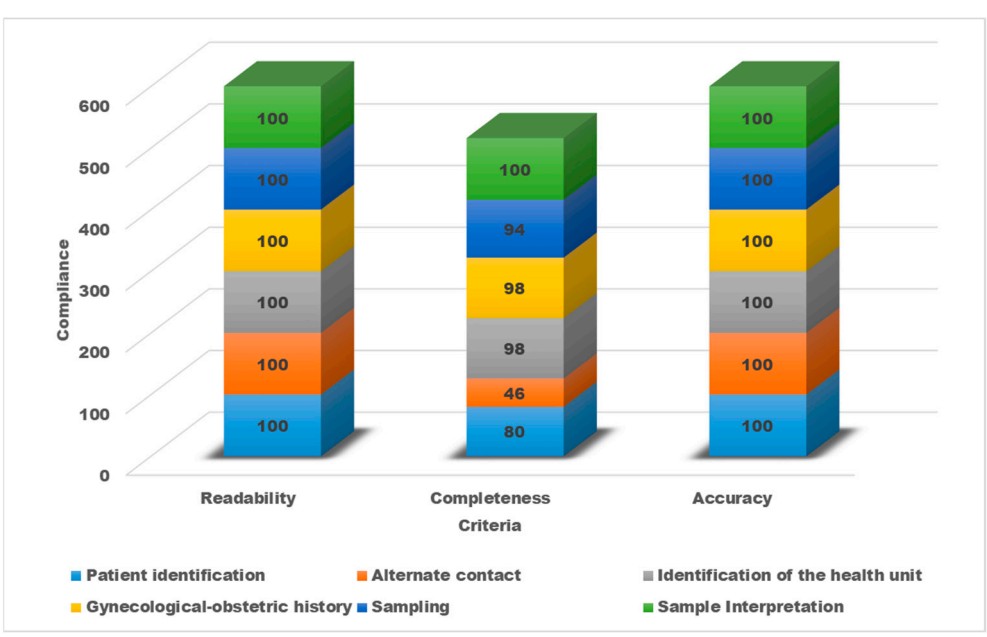

*n=50*

**Figure 4.** Percentage distribution of compliance of the clinical data record with the DCS-TDCC.

## 4. Discussion

Electronic health records can generate quality information and facilitate the exchange of data to help reduce the fragmentation of health systems and health programs, Kruse et al. (2018) point out that health services must join forces so that Health Information Systems have a" meaningful use" to improve the quality and efficiency of health care and reduce population health inequalities; this will facilitate the availability, accuracy, and completeness of data [25]. On the other hand, Cifuentes et al. (2015) mention that some of the challenges that arise in Health Information Systems are related to the duplication of data, input and output of information, inaccessible data, scanning and transport of documents,

and the use of patient monitoring systems [26]. Another weakness that has been reported in the scientific literature is that there is a limited quality of nursing documentation and the need to increase knowledge and skills in these processes, as well as in the application of both paper and electronic systems [27].

In this sense, the CCTDP that is a priority requires a tool with the potential to improve the quality of care. The DCS-TDCC is an alternative for health services where there are no information systems with interoperability and a data network that is organized in such a way that there is continuity in the services provided to patients from their origin. First contact with health personnel in primary care units, as mentioned by Wang et al. (2013) where it is mentioned that electronic nursing documentation systems could improve the quality of the structure and format of the documentation, the process, and the content in the aspects of quantity, completeness, signing, and dating of the evaluation forms [28].

In this study, the evaluation of the structure was favorable since the system complied with all the Official Mexican Standards for the establishment of an Information System and for the creation of electronic clinical records by having all the indicators indicated for it [13,23,24]. The system has an organized and clear structure compared to the official format of health services nationwide, in addition to having as a specific feature of the interconnection between departments and institutions, an essential component for the success of an electronic health record information system. Kuziemsky et al. (2017) refer in this regard that the premises of the HIS are connectivity and socialization, as well as the interoperability that implies the use of standards for data exchange and terminology for their development [29].

As for the registration process, the system allowed for the generation of readable and accurate data in all the records made in the form of digital capture of the data. However, variability of completeness was found because the patients did not have enough data to provide all the required information. Some authors point out that erroneous or incomplete data is produced by socio-cultural barriers between health personnel and users of health services [30]. Therefore, this variability in the completeness of the data is not attributed to the system since the DCS-TDCC has mandatory fields and fields with predefined data for registration, which facilitates the process of communication and interoperability [31,32], and thus avoids that there are imprecise diagnoses with false positives or false negatives in the interpretation of cervical cytology by the pathologist [18,33].

An important finding during the implementation of the project was the detection of areas of opportunity in the abilities of health personnel to obtain information during the interrogation, which implies great institutional challenges for the adoption of policies that guarantee the transfer of new technologies. Therefore, it is desirable to provide continuous training to human resources in health for the use of electronic tools by registration regulations, as well as in the cervical screening process from the pre-analytical, analytical, and post-analytical phases [30,34].

Initially, health professionals showed difficulty in the transition from paper documentation to documentation in digital form; however, later, and with the support of the training provided, it was possible to adopt the system daily in your work area with benefits in inter-institutional communication, such as the decrease in the time used for registration, the sending of applications, and the delivery of results.

In the result dimension, users of the DCS-TDCC perceived ease and speed for electronic registration, with clearer and more complete data than with the manual format, optimization of time for their core activities, a notable decrease in manual documents for the integration of information, as well as with accuracy and rapidity in the delivery of the diagnosis issued by the pathologist, due to the organized structure of the electronic system concerning the official format [35].

Some authors, such as Wagner, have also considered the evaluation of ergonomic criteria (organization, content, interface, and technique) to reduce the gap between health personnel and technology with favorable results in the technical aspect [36]. The foregoing

highlights the importance of attending to health information standards and needs as a reference for the design or restructuring of electronic systems.

The use of the electronic system brought benefits to the program, the institution, health personnel, and users, among which are: ease in registering patient information, optimization of time, and optimization of material resources such as paper, ink, and pens; the professionals highlight the orderly structure of the data and greater information security [37]. Precise diagnoses with timely delivery and ease of consulting patient information were possible since information regarding the process of care for the timely detection of CC could be accessed.

With this, users expressed the desire to continue using the system in each of the departments in which it was implemented. They even mentioned that it is the best option to make patient records since it avoids the risk of loss of files and contributes to the unification of systems with homogeneous and interoperable information. For the pathologist, the system was friendly and easy to use, which makes it possible to give agility to the report of results.

The DCS-TDCC is a tool that contributes to the innovation and strengthening of health systems, as well as in compliance with official Mexican norms of CC and health information systems, in addition to promoting environmental sustainability. It is important to consolidate electronic systems developed based on health information needs, with adequate infrastructure, support, and the necessary resources for their functionality, in such a way that it is possible to contribute to the control and epidemiological surveillance of public health problems, such as CC, to help to reduce the high incidence and mortality rates in the female population.

## 5. Conclusions

The DCS-TDCC is an innovative system that allows the registration of cervical cytology data, with access for all health personnel involved in the screening process: nursing professionals, cytotechnologists, physicians, and pathologists; a feature that allows it to be interoperable between departments and institutions. The system turned out to be very useful for making the work of health personnel at the first level of care more efficient, as well as for systematizing the clinical records of patients and better organizing the screening service. Both health and pathology personnel were satisfied with the use of the system and considered it to be an important tool to improve the quality of the Early Detection of Cervical Cancer program.

The introduction of new technology is a continuous process that requires time and great effort to achieve its proper functionality and is effective for the purposes it has been designed. With the implementation of DCS-TDCC in a relevant context, level 7 of the methodology has complied with the "Technology Readiness Level"; however, limitations were identified in the implementation, such as the reduced training time for health personnel to manage the system, as well as the availability of managers and some health professionals to change the way of working from the manual to the electronic system.

It is recommended to evaluate it through different models to test the ergonomics and usability that it has, likewise, continue with tests to validate it in a wider panorama, for a longer time and with the participation of health personnel from different institutions until reaching TRL 8, to apply to the program of CCTDP successfully and possible its commercialization (TRL 9).

It is suggested to measure the impact generated using the electronic system through the comparison of the co-conventional process and the innovation of the process for the interpretation and reporting of cervical cytology results.

**Author Contributions:** S.O.G.-E. conceptualization, design, and supervision of the project. M.C.G.-Z.: project implementation. C.O.-O.: validation and pathology analysis. Y.T.-F.: structure and preparation of the original draft. J.M.A.-A.: software design and validation. All authors have read and agreed to the published version of the manuscript.

**Funding:** This research received no external funding.

**Institutional Review Board Statement:** The study was conducted in accordance with the Declaration of Helsinki. The project was approved by the Research and Ethics Committee of a "Women and children hospital", with registration number: HNM/03-2015-024, 28 April 2015.

**Informed Consent Statement:** All participants signed an informed consent letter.

**Data Availability Statement:** The data (anonymized, with no identifying information) are available upon reasonable request to the corresponding author.

**Conflicts of Interest:** The authors declare that there is no conflict of interest in this work.

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
