# Peer review of "Computer System for the Capture and Preparation of Cytopathological Reports for Cervical Cancer Detection and His Utility in Training for Health Personnel"

_ejihpe, doi:10.3390/ejihpe12090092_

Round 1

Reviewer 1 Report

The present paper describes the results of the implementation of technological innovations in the process of the capture and preparation of cytopathological reports. An electronic system was structured based on national standards regarding cervical cancer control.  The system was evaluated based on the indicators of structure, process, and result. The system was effective and yielded readable and accurate data, making the process of information capture and delivery of cervical screening results, faster and more efficient.

This article is not a usual research paper, but a presentation of a system that facilitates the process of the capture and preparation of cytopathological reports. Thus, the review is based merely on the value of the content and the quality of the presentation, given that it is suitable for the journal or the SI.

 The paper is well written and in general, it is comprehensible, providing the right information.

 There are some points, however, that could be addressed to improve the paper.

-In the introduction section, please make more explicitly, or summarize the needs for developing such a system.

 - Are there similar systems, at the international level? What are the different and innovative elements of the present system?

-The quantitative evaluation is not adequately presented and it needs, more information and data analysis to be provided. (as it is reported in “data processing’ section).

 The paper has the potential to make a considerable contribution.

Author Response

Thanks for your comments. Please see the file.

Reviewer 2 Report

This manuscript is written well. 

Please add about the limits of your research.

Author Response

(The authors gave the same response as above.)

Reviewer 3 Report

Comments to Authors              

            This study showed that it is suggested to measure the impact generated using the electronic system through the comparison of the co-conventional process and the innovation of the process for the interpretation and reporting of cervical cytology results.

          Authors are kindly requested to emphasize the current concepts about these issues in the context of recent knowledge and the available literature. This articles should be quoted in the References list.

References

11) Computer-aided diagnosis tool for cervical cancer screening with weakly supervised localization and detection of abnormalities using adaptable and explainable classifier. Med Image Anal. 2021;73:102167. doi:10.1016/j.media.2021.102167.

22) Real-world treatment drop-off among recurrent or metastatic cervical cancer patients: A US community oncology-based analysis [published online ahead of print, 2022 Jul 29]. Gynecol Oncol. 2022;S0090-8258(22)00509-1. doi:10.1016/j.ygyno.2022.07.026.

Author Response

(The authors gave the same response as above.)

Round 2

Reviewer 1 Report

The revised is improved according to the comment,

I endorse Publication.